# Parkinson’s Disease: A Narrative Review on Potential Molecular Mechanisms of Sleep Disturbances, REM Behavior Disorder, and Melatonin

**DOI:** 10.3390/brainsci13060914

**Published:** 2023-06-06

**Authors:** Mohammad-Ali Samizadeh, Hamed Fallah, Mohadeseh Toomarisahzabi, Fereshteh Rezaei, Mehrsa Rahimi-Danesh, Shahin Akhondzadeh, Salar Vaseghi

**Affiliations:** 1Cognitive Neuroscience Lab, Medicinal Plants Research Center, Institute of Medicinal Plants, ACECR, Karaj 3365166571, Iran; 2Department of Basic Sciences, Faculty of Veterinary Medicine, University of Tehran, Tehran 1417935840, Iran; 3Psychiatric Research Center, Roozbeh Psychiatric Hospital, Tehran University of Medical Sciences, Tehran 13337159140, Iran

**Keywords:** Parkinson’s disease (PD), REM behavior disorder (RBD), sleep disturbances, melatonin

## Abstract

Parkinson’s disease (PD) is one of the most common neurodegenerative diseases. There is a wide range of sleep disturbances in patients with PD, such as insomnia and rapid eye movement (REM) sleep behavior disorder (or REM behavior disorder (RBD)). RBD is a sleep disorder in which a patient acts out his/her dreams and includes abnormal behaviors during the REM phase of sleep. On the other hand, melatonin is the principal hormone that is secreted by the pineal gland and significantly modulates the circadian clock and mood state. Furthermore, melatonin has a wide range of regulatory effects and is a safe treatment for sleep disturbances such as RBD in PD. However, the molecular mechanisms of melatonin involved in the treatment or control of RBD are unknown. In this study, we reviewed the pathophysiology of PD and sleep disturbances, including RBD. We also discussed the potential molecular mechanisms of melatonin involved in its therapeutic effect. It was concluded that disruption of crucial neurotransmitter systems that mediate sleep, including norepinephrine, serotonin, dopamine, and GABA, and important neurotransmitter systems that mediate the REM phase, including acetylcholine, serotonin, and norepinephrine, are significantly involved in the induction of sleep disturbances, including RBD in PD. It was also concluded that accumulation of α-synuclein in sleep-related brain regions can disrupt sleep processes and the circadian rhythm. We suggested that new treatment strategies for sleep disturbances in PD may focus on the modulation of α-synuclein aggregation or expression.

## 1. Introduction (Parkinson’s Disease)

Parkinson’s disease (PD) is the second most common neurodegenerative disease after Alzheimer’s disease [1]. In addition, PD is the most common neurodegenerative disorder that affects movements and mood states. It has been estimated that PD affects at least 1% of the population over the age of 60 [2], rising to 1–3% among persons above 80 years [3]. PD has a slow onset but is progressive. Tremor is usually the first symptom of PD, while later can be associated with bradykinesia and rigidity [2]. Unfortunately, and with an aging population, it has been estimated that the prevalence and incidence of PD may increase by more than 30% by 2030 [4]. PD is associated with a high risk of disability that is mostly associated with non-motor symptoms, which are difficult to treat or manage [5]. Bradykinesia, rigidity, and tremor at rest are the principal motor symptoms of PD patients [6]. Bradykinesia occurs in 80% to 90% of patients and is correlated with slowness of movement and decreased amplitude of movement. In addition, rigidity is observed in 80% to 90% of patients with PD that contains resistance to passive movement in both flexor and extensor muscles and is usually correlated with the “cogwheel” phenomenon [6]. Postural instability, dystonia, and dysarthria are the other important motor symptoms of PD [7]. However, PD has a wide range of non-motor symptoms, including neuropsychiatric, autonomic, and sensory symptoms. The neuropsychiatric symptoms consist of anxiety, depression, anhedonia, panic, psychosis, dementia, etc. [8,9,10]. PD autonomic symptoms include constipation, orthostatic hypotension, sexual dysfunction, urinary retention, etc. [11,12]. PD sensory symptoms also include olfactory dysfunction (hyposmia), paresthesia, and pain [6]. In addition, sleep disturbances/disorders are highly observed in patients with PD [13]. For example, it has been shown that 60% of patients with PD experience insomnia [14]. In addition, a previous study evaluated a cohort of 412 patients with PD and showed that 209 of them (51%) had trouble with the initiation of sleep or fragmentation during the 5-year study [15]. Importantly, rapid eye movement (REM) behavior disorder (RBD) is one of the most common sleep disorders observed in PD patients [16,17]. RBD is a parasomnia that is characterized by loss of muscle atonia, and also, abnormal behaviors during the REM phase of sleep, often as dream enactments that may lead to injury [18]. However, before further describing sleep disturbances and RBD, we discuss the pathophysiology of PD.

Note that, according to the PRISMA guidelines, some specific key terms were selected before writing the article. The key terms used in the search strategy were as follows: (Parkinson’s disease[Title/Abstract]) AND (sleep[Title/Abstract]) OR (sleep disturbances[Title/Abstract]) OR (sleep disorders[Title/Abstract]) OR (REM behavior disorder[Title/Abstract]) OR (REM sleep behavior disorder[Title/Abstract]) OR (melatonin[Title/Abstract]); (Melatonin[Title/Abstract]) AND (REM behavior disorder[Title/Abstract]) OR (REM sleep behavior disorder[Title/Abstract]) OR (sleep disturbances[Title/Abstract]) OR (sleep disorders[Title/Abstract]). PubMed and Google Scholar databases were searched. There was no publication date limit for the search strategy; however, recent findings were preferable. In addition, there were no inclusion or exclusion criteria, and both clinical and preclinical studies or review and original studies were considered. In order to select articles, the abstract of each article was evaluated, and if the content was relevant to at least one of our objectives, the full text was reviewed. Furthermore, the search period was from 30th March to 1st May 2023.

## 2. Neuropathology of PD

### 2.1. Spread of α-Synuclein

At first, the role of α-synuclein in PD should be discussed. α-synuclein is enriched in synapses and involved in synaptic vesicle function [19]. The seminal Braak staging system [20] has declared that the initial α-synuclein pathology arises inside the CNS in brain-first PD, likely rostral to the substantia nigra pars compacta and spreads via interconnected structures, which negatively affects the autonomic nervous system [21]. The initial α-synuclein pathology in the brain emanates in the dorsal motor nucleus of the vagus and/or the olfactory bulb, probably induced by an enteric and olfactory epithelial insult [22]. Then, the α-synuclein pathology spreads to the rostral brainstem and supratentorial systems. It seems that damage to peripheral autonomic structures may lead to dysfunction [23]. Of note, it has been reported that REM atonia in animal models is significantly mediated by the sublaterodorsal nucleus and precoeruleus region in the pons, and lesions in these brain areas can lead to RBD [24]. According to these reports, a premotor stage of PD, which originates in the brain structures caudal to the substantia nigra and likely in the peripheral nervous system, can be explained.

On the other hand, there is evidence showing that the Braak staging system can be invalid for some Lewy pathology-positive cases. It has been shown that there is no observed pathology in the dorsal motor nucleus of the vagus, despite an observed pathology in higher Braak stage structures, including the locus coeruleus and substantia nigra [25,26]. In addition, less than half of patients with PD have RBD at the time of PD diagnosis [27], although the RBD prevalence can be increased over time [28]. In addition, RBD patients without PD experience severe autonomic degeneration, but almost half of early PD patients with unknown RBD status show normal cardiac sympathetic innervation [29,30]. Therefore, some studies have shown that PD pathology does not start in regions caudal to the substantia nigra in all PD patients, and initial pathology may start in other brain regions [21].

According to these inconsistent reports, there are two different hypotheses: (1) “Brain-first PD” that declares α-synuclein pathology originates inside the brain and spreads by connected neural areas throughout the brain, eventually involving the autonomic nervous system; (2) “Body-first PD” that declares α-synuclein pathology originates in the enteric nervous system with subsequent dispersion through the autonomic nervous system to the lower brainstem, and then, to the rest of the brain [31]. In the early phase of PD, the two subtypes are different in various clinical and imaging markers, while in later phases of PD, the two brain-first and body-first suggested mechanisms may converge due to increasing amounts and dissemination of α-synuclein pathology [21]. This framework declares that misfolded α-synuclein originates at a single site in the nervous system and then spreads via connected neurons in a rather stereotypical manner [32]. This framework has also suggested that α-synuclein pathology coincides with neural impairment and degeneration in PD, even though the detailed interplay of α-synuclein pathology with other mechanisms has not been investigated. In addition, misfolded and aggregated α-synuclein may have a main role in nigral dopaminergic neuronal loss [21].

### 2.2. Loss of Dopaminergic Neurons

The pathophysiology of PD is complicated. PD negatively affects the extrapyramidal system, which consists of the motor-related regions of the basal ganglia and is characterized by the impairment of the dopaminergic system, leading to diminished motor function and clinical symptoms [33]. The basal ganglia is a cluster of nuclei (the caudate and lenticular nuclei (the putamen, globus pallidus externus or GPx, globus pallidus internus or GPi), the subthalamic nucleus (STN), and the substantia nigra (SN)), placed deep to the neocortex of the brain with a wide range of functions such as mediating reward, memory, and movements [34]. Interestingly, patients with PD experience motor symptoms when 50–80% of dopaminergic neurons have been lost, showing the presence of a compensatory mechanism in the early stages of PD [6]. Although dopaminergic dysfunction has been considered as a main cause of the motor symptoms of PD, however, non-motor symptoms of PD support the involvement of other neurotransmitter systems such as glutamatergic, cholinergic, serotonergic, and adrenergic systems, along with neuromodulators such as adenosine and enkephalins [35,36,37]. In addition, it has been revealed that PD seems to be originated in the dorsal motor nucleus of the vagal and glossopharyngeal nerves and in the anterior olfactory nucleus; thus, PD may begin in the brainstem and ascend to higher cortical levels [20]. In addition, the most important histopathological features of PD are the loss of pigmented dopaminergic neurons and the formation of Lewy bodies [38,39]. Dopaminergic neurodegeneration in PD occurs in the nigrostriatal pathway, which projects from substantia nigra pars compacta (SNpc) to the striatum [6]. Progressive dopaminergic neurodegeneration in the striatum of PD patients leads to enhanced activity in the GPi/substantia nigra pars reticulata (SNpr) circuit and the impairment of gamma aminobutyric acid (GABA) function, leading to the inhibition of the thalamus. In the following, the attenuated ability of the thalamus to activate the frontal cortex leads to reduced motor activity in patients with PD [6]. Importantly, progressive degeneration of dopaminergic neurons in PD does not only decrease the activation of the thalamus, but also increases cholinergic function that is related to the loss of dopamine’s normal inhibitory effect [40,41].

### 2.3. Lewy Pathology

Lewy pathology in PD includes the formation of intracytoplasmic Lewy bodies (LBs) with inclusions mostly containing α-synuclein and ubiquitin and Lewy neurites (LNs), which are the neuronal projections of similar inclusions [42]. LBs are protein inclusions that consist of disaggregated oligomers of various proteins [43]. In addition, α-synuclein is normally enriched in synapses and significantly participates in synaptic vesicle function [19]. Importantly, the formation of LBs has been declared an important feature of neural degeneration due to the presence of a neural loss in the predilection sites for LBs [44]. In PD, LBs have been observed in the dopaminergic neurons of the substantia nigra as round bodies with radiating fibrils [38]. LBs are produced following excessive production of misfolded forms of ubiquitin proteins that are significantly involved in protein recycling. LBs formation seems to have a critical role in PD neurodegeneration, with different lesion patterns observed at different stages of PD. Lesion patterns in the dorsal nucleus, medulla, and pons seem to be related to the early (premotor) olfactory and REM symptoms of PD, whereas lesions in the nigrostriatal pathway in the later stages of PD are associated with the common motor symptoms of the disease [45,46]. Furthermore, LBs are correlated with the dementia of PD, similar to their presence in patients with dementia with LBs (DLB). DLB is the second most common cause of neurodegenerative dementia (15–20% of all dementia cases) after Alzheimer’s disease [47], and more than 80% of patients with PD develop dementia [48]. PD and DLB have differences in some features; for example, motor features in PD are more prominent and occur earlier in comparison with DLB [49,50]. Importantly, sleep disturbances are important clinical features of DLB, including poor subjective sleep quality, excessive daytime sleepiness, and RBD [51]. Sleep disturbances in PD and DLB have deleterious effects on quality of life due to the impairment of cognition, motor functions, and the capacity to manage activities of daily living [52].

## 3. PD and Sleep

### 3.1. Sleep Disturbances in PD

Various brain regions and neurotransmitter systems are critically involved in regulating sleep and the sleep/wake cycle. It is not surprising that many of these are affected in PD patients. In addition to the pathophysiology of PD and its effect on sleep/wake-related brain regions, other factors can affect the regulation of sleep, including dopaminergic drugs, as well as other medications used in PD patients, co-morbidities, and genetic factors [53,54,55]. One of the most important side effects of dopaminergic drugs is impulse control disorders, and also, lifestyle factors have a role in the development and continuation of sleep disturbances in PD patients [55]. It has been revealed that the prevalence of insomnia does not increase at the 5-year follow-up in early PD patients, but after the initiation of dopaminergic medication, disorders of sleep maintenance are significantly increased over time, while disorders of sleep onset are reduced [56]. Dopaminergic medication is significantly related to daytime sleepiness in PD [57] and also to falling asleep while driving [58]. However, there are inconsistent reports. It has been shown that poor sleep quality and changes in sleep architecture in PD patients can be improved by levodopa following the improvement of motor symptoms (reduction of rigidity and tremor) [55]. The effect of dopamine agonists on sleep disorders in PD has been evaluated usually as a secondary endpoint in many randomized controlled trials showing an improvement in sleep parameters [59]. Previous research has shown that once-daily ropinirole prolonged release (as an adjunctive therapy to levodopa) significantly improves nocturnal symptoms in patients with PD who suffer troublesome nocturnal disturbances [60]. In addition, it has been concluded that other pathways besides the dopaminergic pathways can affect the quality of sleep in PD, such as the serotoninergic signaling in the dorsal raphe, the histaminergic signaling in the tuberomammillary nucleus, and the cholinergic signaling in the PPT and LDT [61]. In PD, both the sleep macrostructure (often seen as sleep fragmentation and a relative increase in superficial sleep) [62,63] and sleep microstructure (often seen as impaired integrity of some sleep stages, disturbed sleep spindles and K-complexes, or decreased muscle atonia within the REM phase) [64,65] are affected.

There is a large number of studies reporting a high prevalence of insomnia in PD patients [66,67]. It has been reported that insomnia symptoms, including disruptive sleep and non-restorative sleep, are common in patients with PD. In addition, the inability to sleep is more common as a comorbidity than as a separate sleep issue [68]. It has also been revealed that as PD severity worsens, the frequency of insomnia complaints increases [69]. Nocturnal motor symptoms of PD, including tremors, dystonia, akinesia, and restlessness, are significantly involved in insomnia and usually occur in up to 60% of patients [14]. A previous study has shown that motor fluctuations of tremor and rigidity are critically related to difficulty falling asleep in patients with PD. In addition, immobility in bed related to hypokinesia seems to be associated with increased wakefulness after sleep onset [70]. Additionally, depression and anxiety that are observed in between 30 and 60% of PD patients [71] may be involved in sleep disturbances and insomnia. It has been reported that the presence of poor sleep and mood disturbances have a reciprocal relationship, with the presence of one appearing to worsen the other [72,73]. It has been shown that the total duration of sleep in PD patients has a greater relationship with depression severity than that in healthy individuals, suggesting a synergistic effect of depression and PD on sleep health [74].

On the other hand, PD potently affects the regulation of sleep/wakefulness and the function of the circadian rhythm. A previous study has shown that up to 80% of PD patients suffer from sleep/wakefulness disturbance [66]. Importantly, disruptions in the rhythm of melatonin have been highlighted in PD, while it is the major circadian indicator. Particularly, patients with PD do not have significant daytime-dependent variation in plasma melatonin concentrations [75,76]. Previous studies have shown that the diurnal cortisol secretory profile, specifically its peak, is significantly changed in PD patients [75,77]. It has also been suggested that suppressed immune function, neuroinflammation, and oxidative stress are involved in abnormal circadian rhythm in PD [78].

### 3.2. Neural Mechanisms of Sleep Disturbances in PD

Neural mechanisms involved in sleep disturbances in PD [79,80], and even in sleep disturbance itself [81,82], are so complicated. Neurodegeneration in PD (and also other neurodegenerative disorders) directly affects sleep/wakefulness mechanisms and leads to sleep disturbance [83]. PD highly damages the important brain neurotransmitter systems mediating sleep, including norepinephrine, serotonin, dopamine, and GABA [84]. In addition, brain neurotransmitter systems involved in the REM phase, including acetylcholine, serotonin, and norepinephrine, are significantly disrupted in PD [85]. Importantly, although dopaminergic therapy can improve sleep in patients who have nighttime motor impairment, it also negatively affects normal sleep architecture and can be stimulating to some patients [85]. Dopamine promotes waking in different animal models, such as drosophila and mouse [86,87]. It has been suggested that the destruction of the dopaminergic system in PD affects the control of waking, which leads to excessive daytime sleepiness. Therefore, the reduction of dopamine in PD may be responsible for excessive daytime napping due to the role of dopamine in mediating waking and arousal [88]. However, dopamine is significantly involved in mediating REM sleep, and evidence shows a pattern of burst firing of dopaminergic signaling in the REM phase of sleep [89,90]. Although this is an inconsistent role, some hypotheses have been provided. It is possible that various dopaminergic receptors can mediate waking and sleep. Furthermore, dopamine may act to mediate both waking and sleep depending on the concentration of the neurotransmitter [88]. In other words, various levels of dopamine can activate different types of receptors with respect to receptor affinity. In addition, dopamine shows this dualistic role in other physiological phenomena [91,92].

A previous study has shown that the accumulation of α-synuclein in the brain areas responsible for the regulation of sleep is significantly related to impaired sleep and circadian rhythm in patients with PD [88]. There is a higher level of α-synuclein burden in particular sleep-related brain areas in patients with PD suffering sleep disturbance in comparison with PD patients without sleep disturbance [93]. The important sleep-related brain regions include the locus coeruleus and the raphe nuclei that are involved in regulating the REM phase, non-REM sleep, and the transition between them; the posterior hypothalamus, which is related to somnolence; and the thalamus, which is critical in regulating the sleep/wakefulness cycle [24,94]. In transgenic MitoPark mice with slow and progressive degeneration of midbrain dopaminergic neurons, no defects in circadian parameters prior to 19 weeks of age have been shown [95]. After this age, dopamine concentrations are significantly reduced in the midbrain, eventually declining to nearly 3% of their initial level, and all circadian parameters are significantly worsened. Of note, the dopaminergic transmission sends the circadian signal (that originated in the suprachiasmatic nucleus (SCN)) to downstream target neuronal networks, and this role may be an explanation for this result [95]. In sum, α-synuclein deposition and loss of dopaminergic neurons have a crucial role in PD-related pathological changes, and both are contributing factors to the disruption of the sleep cycle and the circadian rhythm [79].

In addition, aging is one of the most principal factors related to the disruption of sleep in PD patients [96]. A substantial age-related reduction in the capacity of the glymphatic clearance system has been reported in mice [97]. An inverse relationship between EEG delta band and glymphatic clearance has also been shown in mice [98]. In addition, aging is related to a decreased EEG delta power [99]. These reports suggest that the reliance of glymphatic function on the slow-wave activity at night clearly shows an important effect of aging on glymphatic clearance. Since age is a principal risk factor for PD, it is possible that age-associated compromises in sleep quality do not only activate but also reduce the manifesting threshold for those disrupted sleep mechanisms particularly associated with the pathogenesis of PD [96].

Melanopsin-containing retinal ganglion cells (mRGCs) are significantly involved in the non-image vision-forming pathways projecting to a wide range of central nervous system (CNS) areas and potently mediate various physiological responses, such as the entrainment of circadian rhythms to the light/dark cycle, acute control of locomotor activity, regulation of sleep, and the pupillary light reflex [100]. mRGCs innervate the SCN, which synchronizes the circadian rhythm, and also the lateral hypothalamus and the ventrolateral preoptic nucleus, which are mainly responsible for the regulation of sleep behaviors [101]. Post-mortem studies have indicated that the retinal melanopsin-positive system is highly degenerated in patients with PD, as revealed by a significantly reduced density and morphological changes of mRGCs in patients with PD in comparison with healthy individuals [102]. Furthermore, the decrease in dopaminergic amacrine cells and the loss of synaptic connections with melanopsin cells in PD may be involved in the degeneration of mRGCs [103], showing the bidirectional relationship between the dopaminergic system and mRGCs. Evidence has revealed that mRGCs are modulated by the dopamine that is released from dopaminergic cell innervations [104,105], whereas mRGCs are involved in the continuous dopaminergic response to light [106]. In addition, there is large evidence reporting the relationship between mRGCs degeneration and sleep disturbances in PD [107,108,109]. Therefore, mRGCs may be involved in the pathophysiology of sleep disturbances in PD (Table 1).

## 4. PD and REM Behavior Disorder (RBD)

### 4.1. RBD Definition

REM behavior disorder (RBD) is one of the most important clinical symptoms of PD and also other neurodegenerative diseases, such as multiple system atrophy (MSA) [110]. RBD is a parasomnia that is characterized by the loss of muscle atonia and abnormal behaviors during the REM phase, often as dream enactments that may lead to injury [18]. Abnormal behaviors or movements observed in the REM phase can be simple, such as muscular twitches, or complex, such as kicking and fighting, which can be dangerous for both the patient and the bed partner. Dreams during the REM phase, which are often intense, obscene, and frightening, can provoke abnormal motor behaviors as a response [111]. It has been reported that the typical RBD dream contents include recurring themes of being chased, attacked, or defending their partner from attack, and behaviors or reactions to these dreams often involve hitting, kicking, or even attempted strangulation with vocalizations such as screaming, shouting, or even laughing [112,113]. A previous meta-analysis study has reported that the prevalence of RBD in PD patients is between 19 and 70% [114]. Importantly, the severity of RBD can precede several years after the onset of the motor features of PD. Thus, idiopathic RBD can be considered a premotor biomarker in PD [36]. Idiopathic RBD significantly increases the risk for the development of neurodegenerative diseases, from 35% at 5 years to 73% at 10 years and to 92% at 14 years [115].

### 4.2. Neuropathology of RBD

#### 4.2.1. RBD-Related Changes in PD Patients

RBD is common among α-synucleinopathy disorders, such as PD or MSA, probably because in α-synucleinopathy disorders, cell loss often occurs in neural structures that regulate REM sleep atonia [113]. RBD is present in 25–58% of patients with PD [113]. A previous meta-analysis study has also shown that nearly half of PD patients are suffering from RBD [114]. Of note, RBD has been considered as an important preclinical marker of PD [116]. The co-occurrence of RBD and PD is important due to the fact that RBD symptoms may precede the onset of the motor symptoms of PD by several years [117]. PD patients with RBD experience an increased disease severity and more accelerated motor progression [118]. It has been declared that Parkinsonian rigidity is affected by the presence of RBD symptoms. Patients with mild or moderate PD and RBD usually show more pronounced and symmetric forearm rigidity [119]. Importantly, RBD is significantly related to some specific PD phenotypes. Age, sex, disease duration, motor disability, dopaminergic medication, motor phenotypes, cognitive functions, and autonomic dysfunctions are different in PD patients with RBD compared to PD patients without RBD. In this regard, it has been shown that RBD in PD has specific phenotypes such as older age, longer disease duration, higher motor dysfunction, higher levodopa doses, hallucination, and cognitive dysfunctions, which may show that RBD in PD may be related to a more extensive and profound pathology or a different underlying pattern of pathologic progression [116]. Furthermore, PD patients with RBD show an altered motor profile [120], a different pattern of neuronal activity on EEG [121], and a unique autonomic presentation [122], compared to PD patients without RBD. It has also been reported that PD patients with RBD have lower scores on neuropsychological tests than those without RBD [123,124]. Particularly, PD patients with RBD have greater levels of cognitive impairments related to learning and executive functioning in comparison with those without RBD [125].

#### 4.2.2. Neural Structures Involved in Regulating REM Sleep Atonia

Previous reports have shown that neural structures that are involved in the regulation of REM sleep atonia are located in the pontine subcoeruleus nucleus and ventral medulla of the brainstem [126,127], along with the amygdala that is involved in the emotional content of dreams [113]. Evidence has shown that these nuclei project inhibitory signals to the cells of the spinal anterior horns, leading to a loss of skeletal, muscular tone [24,128]. In PD, signal intensity in the locus coeruleus/subcoeruleus is significantly attenuated, associated with the loss of muscle atonia during the REM phase [129]. Furthermore, decreased thalamic volume on voxel-based morphometry has been shown in PD patients with RBD [130]. Pathological hyperecogenity of the substantia nigra has also been reported in both idiopathic RBD and PD patients [131]. A previous study has reported a decreased gray matter volume in different brain areas, such as the left posterior cingulate gyrus and the hippocampus, in PD patients with RBD [132]. Importantly, evidence has suggested that RBD is not induced following impairment of the dopaminergic nigrostriatal signaling, and it has not been reported in approximately half of PD patients, and also, the use of dopaminergic medications often cannot improve RBD, while RBD precedes the onset of PD for several years in many patients with PD [115]. Thus, it seems that RBD is induced following a degeneration of the neural system generating muscle atonia during the REM phase.

#### 4.2.3. The Role of GABA and Glycine

Previous studies have shown that decreased glycine and GABA inhibition is responsible for suppressing muscle activity during sleep, leads to RBD, and may share a common pathway with the pathophysiology of PD [133,134]. In addition to the role of dopaminergic loss in the pathophysiology of PD, impairment of GABA and glycine function has also been reported [110,133]. In the pathophysiology of PD, at first, there is a reduction of nondopaminergic cells and then the characteristic dopaminergic loss. Thus, it has been suggested that loss of glycine and GABA inhibition can be considered as initial RBD pathophysiology, whereas progressive dopaminergic deficit may be involved in the worsening of both RBD and PD [133]. Other studies have shown that abnormal glycine and GABA signaling may also be involved because glycine- and GABA-mediated inhibition of skeletal motoneurons is partly involved in suppressing muscle activity during the REM phase of sleep [135,136]. In addition, patients with impaired glycine and GABA transmission usually have heightened motor activity during sleep [137,138]. In addition, medications that augment GABAergic activity, such as clonazepam and melatonin, are the most common and effective drugs for motor-related symptoms of RBD [139,140].

#### 4.2.4. Changes of Norepinephrine, Dopamine, Serotonin, and Acetylcholine in RBD

Importantly, other neurotransmitter systems may be involved in the induction of RBD. A previous study has reported autonomic dysfunction with a somatosensory deficit and reduced norepinephrine level in RBD, which can serve as a possible prodromal marker for developing alpha-synucleinopathy such as PD [141]. Furthermore, a previous post-mortem study has shown severe monoaminergic cell loss in the locus ceruleus (noradrenergic) of an RBD patient [142]. In addition to norepinephrine, most of the studies evaluating the presynaptic dopaminergic terminal have reported a robust reduction in the striatal tracer uptake in many individuals diagnosed, at least initially, with RBD [143,144,145]. It has also been shown that many individuals initially diagnosed with idiopathic RBD and decreased striatal tracer uptake at the presynaptic dopaminergic terminal develop neurodegenerative diseases such as PD [146,147]. In addition, the presence of REM sleep without atonia can show the early involvement of dopaminergic signaling on the medullary structure and/or pontine [148]. Pontine and medullary structures are closely connected to the midbrain dopaminergic pathways and may be affected by an imbalance of dopamine levels [149], leading to RBD. Previous research has concluded that increased electromyography activity during the REM phase may be related to the degeneration of the midbrain dopaminergic system in idiopathic RBD and probably to dopaminergic medications in PD [148].

Evidence has also shown the potential role of serotonin in RBD. The induction of RBD following the use of tricyclic antidepressants [150,151], selective serotonin reuptake inhibitors [152,153], or mixed serotonin-noradrenalin reuptake inhibitors [154,155], shows a potential role of serotonin in the pathophysiology of RBD [156]. A previous study has revealed that antidepressant therapy is related to enhanced REM sleep without atonia, both in patients with and without RBD [157]. Furthermore, it has been suggested that the development of RBD following treatments with antidepressants should be an early signal of underlying neurodegenerative disease [152]. This study [152] also reported that although patients with antidepressant-related RBD show a lower risk of neurodegeneration than patients with “purely-idiopathic” RBD, prodromal neurodegeneration markers are still clearly present. Acetylcholine may also have a role in the induction of RBD. It has been shown that patients with idiopathic RBD show reduced uptake of acetylcholine transporter mostly in different brainstem regions, including reticular formation (RF), pontine coeruleus/subcoeruleus complex, tegmental periaqueductal gray, and mesopontine cholinergic nuclei, all responsible for muscle activity during the REM phase of sleep [158]. The induction of RBD following treatment with rivastigmine (the acetylcholinesterase inhibitor) in a patient diagnosed with Alzheimer’s disease has also been shown [159]. Studies on the postsynaptic acetylcholine terminal have reported a decreased tracer uptake in different cortical regions in idiopathic RBD patients [160,161] and reduced neocortical, limbic cortical, and thalamic tracer uptake in PD patients with RBD compared with those without RBD [162].

#### 4.2.5. Circadian Rhythm and RBD

In addition, it seems that circadian rhythm can modulate the pathophysiology of RBD. In healthy individuals, the amount of REM sleep elevates overnight (particularly in the second part of the night); however, it has been shown that this pattern is not characteristic of patients with RBD [163]. Patients with RBD often lost the clock-dependent increase in rapid eye movements index, probably as a consequence of melatonin dysfunction [163]. Previous research has declared that RBD may be related to altered expression of clock genes and delayed melatonin secretion [164]. As we know, melatonin has a crucial role in the modulation of the circadian rhythm and sleep/wake cycle. Therefore, melatonin can play a role or can affect the pathophysiology of PD and/or RBD (Table 2).

#### 4.2.6. Changes of REM-Related Brain Regions in RBD

Before discussing melatonin, it is important to note that reliable hypotheses have focused on the impaired function of REM-related brain regions in RBD. The critical structures in the brainstem involved in the regulation of the REM phase consist of the “REM-off” and the “REM-on” regions. REM-off regions, including the ventrolateral part of the periaqueductal gray matter (vlPAG) and lateral pontine tegmentum (LPT), and the REM-on regions, including the pre-coeruleus (PC) and sublaterodorsal nucleus (SLD), as well as the extended part of the ventrolateral preoptic nucleus (eVLPO), locus coeruleus (LC), laterodorsal tegmental nucleus (LDTN), pedunculopontine nucleus (PPN), and raphe nucleus (RN) [112]. In addition, some brainstem regions have been considered in RBD pathophysiology, including the medullary magnocellular reticular formation (MCRF), locus coeruleus/subcoeruleus complex, pedunculopontine tegmental nucleus (PPT), laterodorsal tegmental nucleus (LDT), and possibly SN [112].

Within the REM phase, excitatory sublateral dorsal/subcoeruleus nucleus glutamatergic neurons augment spinal cord inhibitory interneurons to hyperpolarize and then prevent the spinal motoneuron pool, leading to REM sleep atonia, so sublateral dorsal/subcoeruleus nucleus lesions resulting from brain lesions in the dorsal pons cause REM sleep atonia loss and may lead to clinical RBD [24,157]. Further, posterior hypothalamic hypocretin can further stabilize the REM-active and REM-inactive centers, and in the context of hypocretin deficiency, as in narcolepsy type 1, RBD may also occur [165]. In addition, it seems that activity changes in the LDT and the PPT may underlie the pathophysiology of RBD [166]. Cholinergic neurons of the LDT and PPT regulate arousal, maintain waking, and modulate REM sleep initiation [167], while in PD, the monomeric form of α-synuclein causes excitability and elevates calcium influx, leading to excitotoxicity and damaging the sleep-control function of LDT and PPT [168].

A previous systematic review has concluded that the presence of RBD in PD patients is related to structural and functional changes in many brain areas, mostly in the brainstem, limbic structures, frontotemporal cortex, and basal ganglia [169]. An imaging study has also shown a robust increase in regional homogeneity in the left cerebellum, the right middle occipital region, and the left middle temporal region, and reduced regional homogeneity in the left middle frontal region in PD patients with RBD in comparison with those without RBD [170]. RBD in early PD seems to be related to presynaptic dopaminergic denervation in the ventral striatum and the anterior caudate nucleus [171]. A decrease in the volume of temporal lobes in PD patients with RBD has been shown in many studies [132,172,173]. Furthermore, there is a neural loss in the pontomesencephalic tegmentum (PMT) in RBD [174]. A previous imaging study has shown that, at the subcortical level, there are smaller volumes in the brainstem (PMT and medullary RF), cerebellum, diencephalon (hypothalamus and thalamus), striatum (putamen), and limbic system (amygdala), while at the cortical level, distinct patterns of both smaller (anterior cingulate) and greater (olfactory) gray matter volumes have been shown [175]. Loss of orexin neurons observed in PD due to the hypothalamic volume loss may also be related to increased sleepiness in RBD, and with the reported appearance in some RBD patients of narcolepsy with cataplexy [176].

Lesions in the pons may develop some synucleinopathies, such as PD and RBD [112]. Within the REM phase of sleep, nuclei from the pons activate neurons in the medulla that send descending inhibitory signals to spinal alpha motoneurons, leading to hyperpolarization and muscle atonia. The disinhibition of these neurons seems to induce muscle activity within REM sleep [24] as a suggested mechanism in RBD. Furthermore, it has been suggested that RBD pathology begins in the medulla and pons, which may be related to initial RBD induction in idiopathic RBD. Then, with ascending Lewy disease to the substantia nigra, the motor symptoms of PD begin [46] (Table 3).

## 5. Melatonin

### 5.1. Introduction

In 1958, Aaron Lerner discovered melatonin and isolated it from the bovine pineal [178]. Melatonin is the principal hormone that is secreted by the pineal gland, although many other organs can release melatonin, such as the retina, bone marrow cells, platelets, skin, or lymphocytes [179,180]. As we know, darkness increases melatonin synthesis and secretion, whereas light suppresses it [181]. The pineal gland receives luminous information comes from the retina through the SCN [182]. In humans, melatonin secretion begins after sundown, shows a peak in the middle of the night (2–4 a.m.), and reduces gradually within the second half of the night [183]. A significant increase in sleep propensity at night is often observed two hours after the onset of endogenous melatonin production in humans [184].

In diurnal species, including humans, melatonin affects the SCN to reduce the wake-promoting signal of the circadian clock, leading to the induction of sleep [185]. Furthermore, melatonin affects the default mode network (DMN) areas in the brain to induce fatigue and sleep-like changes in the activation of the precuneus [186,187]. Melatonin, consumed in the afternoon, significantly reduces the activation of the precuneus in healthy young individuals, which is associated with subjective measurements of fatigue [186]. Melatonin induces a sharp reduction in task-related activation in the rosto-medial part of the occipital cortex, which is associated with increased fatigue and sleepiness, while it does not modulate the activity of more caudal and lateral parts of the occipital cortex, and the parietal cortex and the thalamus [187]. Aging, certain diseases such as primary degeneration of the autonomic nervous system and diabetic neuropathy, and some medications, such as β-blockers, clonidine, naloxone, and non-steroidal anti-inflammatory drugs, may disrupt the nocturnal production of melatonin and can be related to impaired sleep [188].

Melatonin has two receptors named MT1 and MT2, which are G protein-coupled receptors [189]. Both MT1 and MT2 receptors are well distributed in the whole brain [190]. It has been reported that the MT1 receptor is expressed in few structures (notably including the suprachiasmatic nucleus and pars tuberalis), while the MT2 receptor is identified in various brain areas, including the olfactory bulb, forebrain, hippocampus, amygdala, and superior colliculus [191]. In addition, co-expression of MT1 and MT2 receptors has not been shown in a wide range of brain regions, and even within the same region, they are rarely present in the same individual cell [191]. Activation of MT1 or MT2 activates G_i_ proteins, prevents adenylyl cyclase, and reduces intracellular cAMP levels [192]. Both MT1 and MT2 receptors mediate the role of melatonin in regulating the circadian rhythm, sleep–wakefulness cycle, neuroendocrine function, and body temperature [193,194,195].

In addition, there is another receptor that has been cloned in different species, including humans. This orphan receptor, which is named GPR50, does not bind melatonin, and its endogenous ligand is still unknown [196]. GPR50 is highly expressed in the hippocampus, hypothalamus, adrenal glands, and pituitary gland [197]. GPR50 was cloned in 1996 and classified as a member of the melatonin receptor subfamily due to its high homology (45%) with MT1 and MT2 at the amino-acid level and due to the presence of characteristic signatures of this subfamily [198]. GPR50 plays a significant role in the pathophysiology of some psychiatric disorders, such as schizophrenia, depression, and bipolar [199,200]. It has been shown that endogenous GPR50 expression levels can significantly regulate MT1 function. In addition, the potential inhibitory effect of GPR50 on MT1 receptor activity has been shown in previous research [201]. GPR50 behaves as an antagonist of the MT1 receptor, which suggests new pharmacological perspectives for GPR50 [196]. On the other hand, it has been shown that MT1 receptor expression is significantly reduced in both the substantia nigra and the amygdala of patients with PD [202]. Another study has shown a reduction in the expression of MT1 and MT2 receptors in the substantia nigra of PD patients [203]. Melatonin shows its hypnotic and chronobiotic effects by affecting MT1 and MT2 receptors located in the substantia nigra. In addition, ramelteon, as a novel melatonin receptor agonist, acts on MT1 and MT2 receptors with a longer duration of action than melatonin [204]. Therefore, although the role of GPR50 in PD is unknown, but it can be involved in the pathophysiology of PD via its effect on MT1 and MT2 receptors. Evidence in this field is so limited, and further research is needed.

### 5.2. Altered Melatonin Level in PD and Its Effect on Sleep Health

There are impairments in melatonin synthesis and irregularities in the production of melatonin in PD [205]. It seems that there is a significant relationship between CSF melatonin levels and the motor symptoms of PD [206]. A past study has reported that circadian secretion patterns of l-dopa-treated PD patients and the control group are so similar, except for a phase advance of the nocturnal melatonin elevation in the PD group [207]. Although some studies have not found a significant difference in the level of melatonin secreted nor in the amplitude of the melatonin rhythm between PD patients and healthy individuals [207,208]. Evidence has considered PD as an endocrine disorder of melatonin overproduction [209], while it has also been reported that melatonin serum level is decreased in patients with PD suffering excessive daytime sleepiness [210]. In PD patients, there are changes in melatonin levels and a reduction in the volume of the hypothalamus [211]. Hypothalamic volume loss may be responsible for the decreased melatonin output [211]. It has been shown that decreased melatonin output in PD is related to altered sleep architecture, including reduced slow-wave sleep (SWS) and REM sleep [75] and excessive daytime sleepiness [210]. A significant reduction in melatonin concentration in early-stage PD patients has also been shown [75]. A previous study has shown that serum melatonin level at two different time points (00:00 and 05:00) during night sleep is lower in PD patients than the controls [212]. Furthermore, there is an earlier nocturnal melatonin peak in patients with PD on levodopa than in the controls [76,208]. Previous post-mortem study has reported that there is a significant decrease in the MT1 and MT2 receptor expression in both the substantia nigra and the amygdala in PD brains [202]. Both melatonin receptors are observed in various brain areas such as the SCN [185], cerebellum [213], hippocampus [214], and central dopaminergic pathways, including the SN, caudate putamen, ventral tegmental areas, and nucleus accumbens [215]. Previous preclinical research has shown fluctuations in serum melatonin levels in rats that are associated with variations in motor function and are attributed to the interaction of monoamines with melatonin in the striatal complex [216]. The down-regulation of melatonin receptors in the brain areas affected by PD suggests the relationship between melatonin and PD [217]. It has been shown that loss or damage of the neurons in the substantia nigra and other brain regions involved in circadian timing may be related to circadian rhythm abnormalities and sleep disturbance in PD [218]. PD patients have blunted circadian rhythms of melatonin secretion, and studies have shown that the amplitude of the melatonin rhythm and the 24 h area under the curve for circulating melatonin level are remarkably lower in PD patients, indicating circadian impairment may underlie excessive daytime sleepiness in PD [210].

It is important to note that endogenous melatonin level is elevated along with increasing dopamine degeneration in a 6-hydroxydopamine-induced rat model [219]. Considering that PD is characterized by irreversible degenerate dopaminergic neurons, melatonin can play a significant role in exerting a neuroprotective effect and, thus, increase compensation [220]. Evidence has shown an increased plasma melatonin level in patients with PD compared to healthy individuals [220,221]. In addition, PD patients have a significant negative correlation between melatonin levels and equivalent levodopa doses, showing that increased plasma melatonin levels in PD patients can be associated with the degeneration of dopaminergic neurons and the use of dopaminergic medications [220]. Dopaminergic treatments may lead to an increase in plasma melatonin levels. The secretion of melatonin in PD patients with l-dopa-related motor complications is significantly increased during the day and decreased at night because dopamine has a critical role in the maintenance of circadian system rhythmicity [222,223]. Thus, elevated melatonin levels and the correlation with the equivalent dose of levodopa in patients with PD may be associated with the disruption of the circadian rhythm and with the imbalance of dopaminergic regulation [220].

With respect to the role of melatonin in mediating the sleep–wakefulness cycle, disrupted melatonin function in PD patients is significantly related to impaired sleep architecture, including decreased slow-wave and REM sleep [75] and excessive daytime sleepiness [210]. Therefore, melatonin therapy may be useful for the treatment of sleep disorders in PD. Previous reports have revealed that melatonin can improve subjective sleep quality in patients with PD [224,225]. Melatonin treatment improves daytime sleepiness and sleep quantity in patients with PD [226]. In addition, melatonin (50 mg) significantly improves total sleep time [226]. It has been shown that the effect of melatonin on daytime sleepiness in PD patients can be different, maybe due to the use of different scales [224]. In addition, melatonin may be useful for the treatment of RBD (Table 4).

## 6. Melatonin and RBD

### 6.1. Evidence

There are published reports indicating the therapeutic role of melatonin in RBD. A previous study has shown that melatonin can be used for the treatment of RBD in PD patients [227]. This study [227] showed that melatonin (3–6 mg for 4 weeks) reduced RBD symptoms and daytime sleepiness in 84% of patients with PD. There are other clinical studies that show melatonin 3–12 mg at bedtime is effective for the treatment of RBD [228,229,230,231]. Furthermore, ramelteon, a novel melatonin receptor agonist, may have therapeutic potential in the treatment of sleep issues such as RBD in patients with PD [203]. Tasimelteon, another MT1/MT2 agonist, may also be effective for sleep resynchronization [232]. A previous trial study has also shown that melatonin is effective in the treatment of RBD [233]. However, there are inconsistent results showing that melatonin has no therapeutic effect on RBD in PD [234,235]. For example, a recent clinical trial has shown that clonazepam but not prolonged-release melatonin improves RBD symptoms [236]. It should be noted that evidence on the effect of melatonin on the treatment of RBD is not enough.

### 6.2. Melatonin’s Suggested Mechanisms

The molecular mechanisms of melatonin against RBD still remain unclear. A previous study has suggested that melatonin’s effect may be mediated by a combination of influences, including a direct impact on REM sleep atonia, stabilizing circadian clock variability and desynchronization, and increasing sleep efficiency [237]. Melatonin significantly reduces the tonic but not the phasic electromyographic activity in the submentalis muscle in patients with RBD [115]. Previous research has reported that melatonin significantly decreases REM sleep without atonia [233]. In addition, the modulatory role of melatonin on GABAergic signaling may be involved in its therapeutic effect. Melatonin significantly potentiates GABAergic function [238]. Previous research has shown that melatonin has more efficacy than clonazepam in decreasing REM motor behaviors and restoring REM muscle atonia in a glycine/GABA-A receptor knock-out transgenic mouse model of RBD [133]. It has been reported that 2–4-week melatonin treatment reduces masseter muscle tone in REM sleep by 43% in transgenic mice [133]. Furthermore, melatonin augments the action of GABA on GABA-A receptors located on motoneurons and may directly potentiate tonic GABA-A transmission at the motoneurons level to decrease muscle tone [239]. Moreover, melatonin significantly decreases the frequency and severity of dream enactment behavior in RBD [240], which can attenuate the severity of RBD symptoms. It has been revealed that melatonin significantly decreases nights with dream-acting-out, nights with vocalizations, and percent of high EMG during total REM sleep time in RBD patients [241].

RBD is closely related to oxidative stress-related damages in PD [217]. In patients with PD, degenerative or dead neurons can release more α-synuclein oligomers into the extracellular space of RBD-related brain regions that can induce persistent microglial activation and neural death. In the following, it can worsen the inflammation in the central and peripheral systems and finally leads to RBD [217]. A previous case report has shown that a brainstem inflammatory lesion leads to RBD [242]. In addition, in RBD, there is elevated microglial activation in the substantia nigra, as well as decreased dopaminergic activity in the putamen [243], suggesting a crucial role of neuroinflammation in the pathophysiology of RBD. A previous study has shown that serum allantoin and allantoin/uric acid ratio are elevated in RBD patients, showing an increased systemic oxidative stress in prodromal synucleinopathy [244]. On the other hand, melatonin exerts anti-oxidative and anti-inflammatory effects [245,246], leading to a therapeutic effect in RBD. A previous study has reported that the administration of melatonin prevents some pathways related to apoptosis, autophagy, oxidative stress, inflammation, α-synuclein aggregation, and dopamine loss in PD [247]. Administration of melatonin also increases superoxide dismutase (SOD), mitochondrial complex-I activity, and glutathione (GSH) in the substantia nigra in a rat model of PD [248].

In addition, patients with RBD taking acetylcholinesterase inhibitors have both lessened response in RBD outcomes [228] as well as an improvement [249]. Melatonin suppresses calmodulin, which then may modulate skeletal muscle acetylcholine receptors [250]. It has been reported that melatonin, through this mechanism, can be important for receptor maintenance in aging persons [251]. It is also presumed that melatonin modulates cytoskeletal structure through its antagonism with calmodulin [252]. The role of melatonin in decreasing calmodulin can subsequently modulate the cytoskeletal structure and nicotinic acetylcholine receptor expression in skeletal muscle cells [237]. Thus, melatonin’s activity as a calmodulin antagonist may impact RBD pathophysiology [237].

The sublaterodorsal nucleus (SLD) and/or magnocellular reticular formation (MCRF) nucleus are presumed to undergo neuronal loss in RBD, such that their projections on the anterior horn cell (AHC) ultimately have decreased effects [237]. It has been revealed that melatonin may reduce the electrophysiologic and behavioral features of RBD via potentiating the action of GABA on GABA-A receptors in the AHC. Melatonin, via normalizing the circadian rhythm, may alleviate RBD symptoms. It has been shown that melatonin increases sleep efficiency and shortens sleep latency with evening administration [253,254], and may stabilize circadian clock variability [255]. Internal circadian rhythm desynchronization may occur in RBD, while melatonin may act to re-entrain the suprachiasmatic nucleus, thereby restoring normal circadian REM sleep modulation [256]. Previous research has reported that agomelatine prior to bedtime reduces the frequency and severity of RBD episodes and reduces dream production over a period of 6-month follow-up, maybe via normalizing REM sleep and increasing dopamine and noradrenaline activity in the frontal cortex, which reduces excitatory glutamate release [257]. A previous study has shown that melatonin can normalize the circadian timing of REM sleep, leading to a decrease in RBD behaviors [256]. Furthermore, melatonin, via affecting tonic REM sleep and restoring normal muscle atonia, can directly influence the pathophysiology of RBD [230] (Table 5).

## 7. Conclusions

In this review study, we aimed to discuss the molecular studies underlying sleep disturbances, particularly RBD in PD. In addition, we reviewed melatonin changes in PD and its potential therapeutic role in RBD. It was concluded that disruption of important neurotransmitter systems mediating sleep, including norepinephrine, serotonin, dopamine, and GABA, and important neurotransmitter systems mediating the REM phase, including acetylcholine, serotonin, and norepinephrine, are significantly involved in the induction of sleep disturbances in PD. Furthermore, the destruction of dopaminergic signaling in PD affects the control of wakefulness and leads to increased excessive daytime sleepiness, while dopaminergic therapy may disrupt normal sleep architecture in PD patients. It was also concluded that accumulation of α-synuclein in the brain regions involved in modulating sleep can disrupt sleep and circadian rhythm. It can be suggested that treatment strategies that involve small molecules with the potential to change the conformation of α-synuclein and render them nonpathogenic would be beneficial. For example, polyphenols interact with α-synuclein aggregates and can destabilize them, leading to a potential therapeutic effect. In addition, various treatment strategies, such as gene therapy and cellular transplantation, with the potential to modulate the expression and aggregation of α-synuclein, may be beneficial.

RBD, as one of the most important sleep disorders in PD, may occur following cell loss within neural structures involved in the regulation of REM sleep atonia. It was concluded that the loss of muscle atonia during the REM phase in RBD may be related to attenuated signal intensity in the locus coeruleus/subcoeruleus. Furthermore, reduced glycine and GABA inhibition, which normally suppresses muscle activity during sleep, is involved in the pathophysiology of RBD. In addition, decreased uptake of acetylcholine transporter in brainstem regions involved in muscle activity during REM sleep, including reticular formation, pontine coeruleus/subcoeruleus complex, tegmental periaqueductal gray, and mesopontine cholinergic nuclei, is significantly related to RBD. Therefore, it seems that GABA, glycine, and acetylcholine are the most important neurotransmitters that are involved in the pathophysiology of RBD. Future studies can focus on the potential role of these neurotransmitters in the treatment of RBD. In addition, changes in the level or function of GABA, glycine, or acetylcholine in RBD should be investigated further.

In RBD, the function of many brain regions involved in modulating the REM phase is disrupted. We declared that sublateral dorsal/subcoeruleus nucleus lesions in the dorsal pons may lead to REM sleep atonia loss. It was concluded that the monomeric form of α-synuclein induces excitability and increases calcium influx, leading to excitotoxicity and damaging the sleep-control function of LDT and PPT. It can be suggested that α-synuclein-related biomarkers can provide a unique chance for the treatment of RBD and PD. We also found that both the decrease and increase in melatonin have been shown in PD. It seems that melatonin, via its antioxidant and anti-inflammatory effects, can induce a therapeutic effect in RBD. Melatonin, via reducing REM motor behaviors and restoring REM muscle atonia, may be beneficial for the treatment of RBD. We also declared that melatonin can attenuate the electrophysiologic and behavioral manifestations of RBD by potentiating the action of GABA on GABA-A receptors in the AHC.

In conclusion, sleep disturbances and PD have a reciprocal relationship with each other. In other words, PD may lead to sleep disturbances, and on the other hand, sleep disturbances may be associated with the progression of PD. Importantly, novel strategies for the treatment of PD should focus on the modulation of α-synuclein. As a limitation, we declare that PubMed and Google Scholar databases were searched in this review study, with a greater focus on the PubMed database. Therefore, other published articles that were not published in these databases were not studied.

## Figures and Tables

**Table 1 brainsci-13-00914-t001:** Mechanisms involved in sleep disturbances in Parkinson’s disease (PD) (mRGCs: melanopsin-containing retinal ganglion cells).

Mechanism Responsible for Sleep Disturbance in PD	Reference
Disruption of important brain neurotransmitter systems mediating sleep, including norepinephrine, serotonin, dopamine, and GABA.	[84]
Disruption of brain neurotransmitter systems involved in REM sleep, including acetylcholine, serotonin, and norepinephrine.	[85]
Brain neurotransmitters involved in sleep regulation (norepinephrine, serotonin, dopamine, and GABA) are variably damaged.	[85]
Destruction of dopaminergic signaling affects the control of wakefulness, leading to increased excessive daytime sleepiness.	[88]
Accumulation of α-synuclein in the brain regions involved in regulating sleep leads to disrupted sleep and circadian rhythm.	[88]
Decrease in the capacity of the glymphatic clearance system induced by aging.	[96,97]
Retinal melanopsin-positive system is degenerated, and decreased density and morphological alterations of mRGCs have been shown.	[102]
The reduction in the number of dopaminergic amacrine cells and the loss of synaptic connections with melanopsin cells are observed.	[103]

**Table 2 brainsci-13-00914-t002:** Suggested mechanisms underlying REM behavior disorder (RBD) pathology.

Suggested Mechanisms Underlying RBD Pathology	Reference
Cell loss occurs within neuronal structures regulating REM sleep atonia.	[113]
Signal intensity in the locus coeruleus/subcoeruleus is significantly reduced, associated with the loss of muscle atonia during the REM phase.	[129]
Decreased thalamic volume on voxel-based morphometry.	[130]
Pathological hyperecogenity of the substantia nigra.	[131]
Diminished gray matter volume in several brain regions, such as the left posterior cingulate gyrus and the hippocampus.	[132]
Decreased glycine and GABA inhibition, the neurotransmitters involved in the suppression of muscle activity during sleep.	[133,134]
Loss of glycine and GABA inhibition may explain the initial RBD pathophysiology, while progressive dopaminergic deficit may explain the worsening of both RBD and PD.	[133]
Impaired glycine and GABA transmission lead to the experience of heightened motor activity during sleep.	[137,138]
Autonomic dysfunction with somatosensory deficit and reduced norepinephrine level.	[141]
Severe monoaminergic cell loss in the locus ceruleus (noradrenergic) of a patient with RBD has been shown.	[142]
Decrease in the striatal dopamine uptake in many RBD patients has been shown.	[143,144]
Antidepressant therapy and serotonergic drugs are related to enhanced REM sleep without atonia, both in patients with and without RBD.	[157]
Reduced uptake of acetylcholine transporter mainly in brainstem regions, including reticular formation, pontine coeruleus/subcoeruleus complex, tegmental periaqueductal gray, and mesopontine cholinergic nuclei, all involved in muscle activity during REM sleep.	[158]
Decreased acetylcholine uptake in different cortical regions.	[160,161]
Decreased neocortical, limbic cortical, and thalamic acetylcholine uptake in PD patients with RBD compared to those without RBD.	[162]
The amount of REM sleep increases overnight in normal conditions; however, this pattern is not characteristic of RBD.	[163]
RBD patients lost the clock-dependent increase in rapid eye movements index, probably as a consequence of melatonin dysfunction.	[163]
Altered expression of clock genes and delayed melatonin secretion.	[164]

**Table 3 brainsci-13-00914-t003:** Changes in melatonin levels in Parkinson’s disease (PD).

Changes of REM-Related Brain Regions in RBD	Reference
Sublateral dorsal/subcoeruleus nucleus lesions resulting from brain lesions in the dorsal pons cause REM sleep atonia loss.	[24,157]
Posterior hypothalamic hypocretin stabilizes the REM-active and REM-inactive centers and networks, while hypocretin deficiency may lead to RBD.	[165]
There are activity changes in the LDT and the PPT.	[166]
Monomeric form of α-synuclein induces excitability and increases calcium influx, leading to excitotoxicity and damaging the sleep-control function of LDT and PPT.	[168]
There are functional changes in many brain regions, mainly in the brainstem, limbic structures, frontotemporal cortex, and basal ganglia.	[169]
A significant increase in regional homogeneity in the left cerebellum, the right middle occipital region, and the left middle temporal region.	[170]
Decreased regional homogeneity in the left middle frontal region.	[170]
Presynaptic dopaminergic denervation in the ventral striatum and the anterior caudate nucleus.	[171]
A decrease in the volume of temporal lobes.	[132,172,173]
Loss of neurons in the region of the PMT.	[174]
Loss of orexin neurons in the hypothalamus.	[176]
Smaller volumes in the brainstem (PMT and medullary RF), cerebellum, diencephalon (hypothalamus and thalamus), striatum (putamen), and limbic system (amygdala).	[175]
Distinct patterns of both smaller (anterior cingulate) and larger (olfactory) gray matter volumes.	[175]
Progressive degeneration of pons nuclei may explain the induction of RBD.	[177]
Disinhibition of the nuclei from the pons, which activate neurons in the medulla that transmit descending inhibitory projections to spinal alpha motor-neurons involved in muscle atonia.	[24]

**Table 4 brainsci-13-00914-t004:** Functional and anatomical changes of REM-related brain regions in REM behavior disorder (RBD) (PPT: pedunculopontine tegmentum; LDT: laterodorsal tegmentum; PMT: pontomesencephalic tegmentum; RF: reticular formation).

Melatonin Level in PD	Reference
There is an earlier nocturnal melatonin peak in patients on levodopa.	[76,208]
There is a significant decrease in the MT1 and MT2 receptor expression in both the substantia nigra and the amygdala.	[202]
Down-regulation of melatonin receptors in the brain regions affected by PD has been shown.	[217]
Melatonin level is related to hypothalamic gray matter volume loss and disease severity.	[211]
Hypothalamic volume loss (also seen in PD) may be responsible for reduced melatonin output.	[211]
PD is an endocrine disorder of melatonin overproduction.	[209]
There is a significant reduction in melatonin concentration in early-stage PD.	[75]
Serum melatonin level at two different time points (00:00 and 05:00) during night sleep is lower in PD patients.	[212]
There is a blunted circadian rhythm of melatonin secretion.	[210]
Both the amplitude of the melatonin rhythm and the 24 h area under the curve for circulating melatonin levels are significantly lower.	[210]
Melatonin serum level is decreased in patients with PD who suffer from excessive daytime sleepiness.	[210]
Endogenous melatonin level is increased along with increasing dopamine degeneration in a 6-hydroxydopamine-induced rat model.	[219]
There is a significant negative correlation between melatonin levels and equivalent levodopa doses, suggesting that the increased plasma melatonin level in PD may be related to the degeneration of dopaminergic neurons and the use of dopaminergic drugs.	[220]
Dopaminergic treatments may lead to an increase in plasma melatonin level.	[222,223]

**Table 5 brainsci-13-00914-t005:** Suggested mechanisms of melatonin in REM behavior disorder (RBD) (AHC: anterior horn cell).

Melatonin’s Mechanisms in RBD	Reference
Exerts anti-oxidative and anti-inflammatory effects.	[245,246]
Prevents some pathways related to apoptosis, autophagy, oxidative stress, inflammation, α-synuclein aggregation, and dopamine loss in PD.	[247]
Decreases the tonic but not the phasic electromyographic activity in the submentalis muscle.	[115]
Decreases REM sleep without atonia.	[233]
Modulatory role of melatonin on GABAergic signaling.	[133]
Significantly potentiates GABAergic function.	[238]
Great efficacy in decreasing REM motor behaviors and restoring REM muscle atonia in a glycine/GABA-A receptor knock-out transgenic mouse model of RBD.	[133]
Potentiates the action of GABA on GABA-A receptors located on motoneurons.	[239]
Potentiates tonic GABA-A transmission at the motoneurons level to decrease muscle tone.	[239]
Reduces masseter muscle tone in REM sleep by 43% in transgenic mice.	[133]
Decreases the frequency and severity of dream enactment behavior.	[240]
Decreases nights with dream-acting-out, nights with vocalizations, and percent of high EMG during total REM sleep time.	[241]
Suppresses calmodulin, which may modulate skeletal muscle acetylcholine receptors.	[250]
Reduces the electrophysiologic and behavioral manifestations of RBD via potentiating the action of GABA on GABA-A receptors in the AHC.	[237]
Reduces calmodulin, which subsequently modulates the cytoskeletal structure and nicotinic acetylcholine receptor expression in skeletal muscle cells.	[237]
Restores REM atonia.	[256]
Normalizes the circadian timing of REM sleep, leading to a decrease in RBD behaviors.	[256]
Stabilizes circadian clock variability.	[255]
Re-entrains the suprachiasmatic nucleus, thereby restoring normal circadian REM sleep modulation.	[256]
Affects tonic REM sleep and restores normal muscle atonia.	[230]

## Data Availability

Not applicable.

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
