# Peer review of "Parkinson’s Disease: A Narrative Review on Potential Molecular Mechanisms of Sleep Disturbances, REM Behavior Disorder, and Melatonin"

_brainsci, 2023, doi:10.3390/brainsci13060914_

Round 1

Reviewer 1 Report

Comments and Suggestions for Authors

The authors adequately review the existing evidence about the cellular/molecular mechanisms mediating sleep disturbances and RBD in Parkinson's disease. This is an important area of neurological research and this review constitutes an excellent refinement of the knowledge of this subject.

Some suggestions to consider,

Neuropathology of PD. Lines 57-86.  In this context it would be important to mention the hypothetical brain first or body firsts mechanistic of spreading of synucleins (Clinical and imaging evidence of brain-first and body-first Parkinson´s disease. Jacob Horsager, Karline Nudsen, Michael Sommerauer, Neurobiology of Disease, Vol. 164, 2022, 105626).

 Melatonin. Lines 375-412. Here it would be interesting to mention the probable role of GPR50 on MT1 and Parkinson’s disease and dyskinetic behaviors.

Comments on the Quality of English Language

No comment

Author Response

The authors adequately review the existing evidence about the cellular/molecular mechanisms mediating sleep disturbances and RBD in Parkinson's disease. This is an important area of neurological research and this review constitutes an excellent refinement of the knowledge of this subject.

Some suggestions to consider,

1) Neuropathology of PD. Lines 57-86.  In this context it would be important to mention the hypothetical brain first or body firsts mechanistic of spreading of synucleins (Clinical and imaging evidence of brain-first and body-first Parkinson´s disease. Jacob Horsager, Karline Nudsen, Michael Sommerauer, Neurobiology of Disease, Vol. 164, 2022, 105626).

Response 1: Thanks for your valuable suggestion. We added a new section: 2.1. Spread of α-synuclein.

2- Melatonin. Lines 375-412. Here it would be interesting to mention the probable role of GPR50 on MT1 and Parkinson’s disease and dyskinetic behaviors.

Response 2: Thanks for your suggestion. In the section 5.1. introduction, we added a new paragraph (last paragraph) and briefly discuss GPR50.

Reviewer 2 Report

Comments and Suggestions for Authors

I reviewed the manuscript entitled Parkinson’s disease and melatonin: A narrative review on potential cellular and molecular mechanisms of sleep disturbances and REM behavior disorder.

The paper is well written and is relevant to the study of  the field of sleep medicine, thus  it will be of interest to the readers of Brain Sciences. The abstract adequatly describes the study. The manuscript is well organized and data are presented in orderly manner.

I found two minor issues:

The introduction is not concise. The melatonin role is discused starting  in page  10. It should be shortened or the title  changed.

Table 1 Authors stated:  "dopaminergic therapy can disrupt normal sleep architecture  . It is true, but it is not mechanism of sleep disruption. Remove it or change.

Author Response

I reviewed the manuscript entitled Parkinson’s disease and melatonin: A narrative review on potential cellular and molecular mechanisms of sleep disturbances and REM behavior disorder.

The paper is well written and is relevant to the study of  the field of sleep medicine, thus  it will be of interest to the readers of Brain Sciences. The abstract adequatly describes the study. The manuscript is well organized and data are presented in orderly manner.

I found two minor issues:

1- The introduction is not concise. The melatonin role is discused starting  in page  10. It should be shortened or the title  changed.

Response 1: Thanks for pointing this out. The main goal of this study is to discuss sleep disturbances in PD. Therefore, as you requested, we changed the title.

2- Table 1 Authors stated:  "dopaminergic therapy can disrupt normal sleep architecture  . It is true, but it is not mechanism of sleep disruption. Remove it or change.

Response 2: Thanks for your precision. This sentence in table 1 was edited.

Reviewer 3 Report

Comments and Suggestions for Authors

The manuscript under review presents a detailed account of molecular studies on sleep disturbances, particularly Rapid Eye Movement (REM) Sleep Behavior Disorder (RBD), in Parkinson's Disease (PD). Although the manuscript is well-drafted overall, there are opportunities for improvement. Here are my recommendations:

Major suggestions:

1.      Abstract Completeness: The abstract could benefit from a more comprehensive summary of the paper's objectives, methodology, and main findings. For instance, it would be beneficial to mention the main neurotransmitters involved in RBD (dopamine, GABA, etc.) and the role of melatonin in the therapeutic context. There is an asymmetry between the Abstract and Conclusions

2.      Background Information: The manuscript could improve by providing more detailed background information about RBD and PD. For instance, the authors could include an overview of the prevalence of RBD in PD patients, how it impacts the quality of life, and the current standard of care for these patients.

3.      Methodology Details: The review methodology needs to be more transparent. It would be beneficial to mention the databases searched, the time frame for the literature review, and any inclusion or exclusion criteria applied to the reviewed articles.

4.      Organization of Findings: The presentation of findings could be improved by organizing the results into subsections according to the different neurotransmitters discussed, such as "Role of Dopamine in RBD in PD," "Importance of Serotonin and Norepinephrine," and "Therapeutic Potential of Melatonin in RBD." This would enhance readability and facilitate comprehension.

5.      Contradictory Findings: The manuscript discusses the role of dopaminergic signaling in PD and how it affects sleep, but it should also address studies that have found contradictory results. For instance, if there are studies that suggest dopaminergic therapy does not disrupt normal sleep architecture in PD patients, these should be included in the discussion.

6.      Implications of Findings: The manuscript could do more to highlight the implications of the findings. For instance, the authors could elaborate on how their conclusions about the role of α-synuclein in sleep disturbances might guide the development of new therapeutic strategies for PD.

7.      Limitations: The authors should acknowledge the limitations of their review. For instance, if the review only included studies published within a certain time frame, for example, this limitation should be stated.

8.      Further Research Recommendations: While the authors recommend further studies on brain regions and the role of melatonin in modulating the function of REM-related brain areas, it would be helpful to provide more specific suggestions. For instance, they could suggest investigations into the use of melatonin supplements in PD patients with RBD, or research into the role of specific neurotransmitters in different stages of PD.

9.   Conclusion Clarity: The conclusion currently states that "sleep disturbances and PD have a reciprocal relation with each other." This statement is a bit ambiguous and could be made clearer. For instance, the authors could specify whether they mean that PD tends to cause sleep disturbances, or that sleep disturbances might accelerate the progression of PD, or both.

Minor suggestions:

10. Interdisciplinary Approach: The manuscript has a strong focus on the biological aspects of PD and RBD, but it could benefit from a broader, interdisciplinary perspective. For example, the psychological and social impacts of these disorders could be discussed, including how they affect quality of life, mental health, and social relationships. This would provide a more comprehensive understanding of the disorders and could also inform treatment and management strategies.

By addressing these points, the manuscript could be significantly enhanced in clarity, comprehensibility, and overall quality.

Comments on the Quality of English Language

There are some awkward sentences that could be reworded for clarity. For instance, as an example, "Dreams within the REM phase are often intense, obscene, and frightening, and the abnormal motor behaviors occur as a response to the contents of the dreams" could be rewritten as "Dreams during the REM phase, which are often intense, obscene, and frightening, can provoke abnormal motor behaviors as a response." 

Author Response

The manuscript under review presents a detailed account of molecular studies on sleep disturbances, particularly Rapid Eye Movement (REM) Sleep Behavior Disorder (RBD), in Parkinson's Disease (PD). Although the manuscript is well-drafted overall, there are opportunities for improvement. Here are my recommendations:

Major suggestions:

  1. Abstract Completeness:The abstract could benefit from a more comprehensive summary of the paper's objectives, methodology, and main findings. For instance, it would be beneficial to mention the main neurotransmitters involved in RBD (dopamine, GABA, etc.) and the role of melatonin in the therapeutic context. There is an asymmetry between the Abstract and Conclusions.

Response 1: Thanks for pointing this out. Abstract was edited.

  1. Background Information:The manuscript could improve by providing more detailed background information about RBD and PD. For instance, the authors could include an overview of the prevalence of RBD in PD patients, how it impacts the quality of life, and the current standard of care for these patients.

Response 2: Thanks for your great suggestion. We added a new section: 4.2.1. RBD-related changes in PD patients, and tried to declare RBD in PD.

  1. Methodology Details:The review methodology needs to be more transparent. It would be beneficial to mention the databases searched, the time frame for the literature review, and any inclusion or exclusion criteria applied to the reviewed articles.

Response 3: Although this is not a systematic review or meta-analysis, we added this info at the end of section 1. Introduction (Parkinson’s disease).

  1. Organization of Findings:The presentation of findings could be improved by organizing the results into subsections according to the different neurotransmitters discussed, such as "Role of Dopamine in RBD in PD," "Importance of Serotonin and Norepinephrine," and "Therapeutic Potential of Melatonin in RBD." This would enhance readability and facilitate comprehension.

Response 4: Thanks for your valuable suggestion. We added some sub-sections to the section 4.2. Neuropathology of RBD, including “RBD-related changes in PD patients”, “Neural structures involved in regulating REM sleep atonia”, “The role of GABA and glycine”, “Changes of norepinephrine, dopamine, serotonin, and acetylcholine in RBD”, “Circadian rhythm and RBD”, and “Changes of REM-related brain regions in RBD”.

  1. Contradictory Findings:The manuscript discusses the role of dopaminergic signaling in PD and how it affects sleep, but it should also address studies that have found contradictory results. For instance, if there are studies that suggest dopaminergic therapy does not disrupt normal sleep architecture in PD patients, these should be included in the discussion.

Response 5: Thanks for your valuable comment. In section 3.1. Sleep disturbances in PD, we added literature in this regard (1st paragraph).

  1. Implications of Findings:The manuscript could do more to highlight the implications of the findings. For instance, the authors could elaborate on how their conclusions about the role of α-synuclein in sleep disturbances might guide the development of new therapeutic strategies for PD.

Response 6: Thanks for your suggestion. In the section: Conclusion, we added literature in this regard (1st, 2nd, and 3rd paragraphs, red lines).

  1. Limitations:The authors should acknowledge the limitations of their review. For instance, if the review only included studies published within a certain time frame, for example, this limitation should be stated.

Response 7: At the end of conclusion, we declared the limitation. Although there was no a certain time frame.

  1. Further Research Recommendations:While the authors recommend further studies on brain regions and the role of melatonin in modulating the function of REM-related brain areas, it would be helpful to provide more specific suggestions. For instance, they could suggest investigations into the use of melatonin supplements in PD patients with RBD, or research into the role of specific neurotransmitters in different stages of PD.

Response 8: Thanks for your valuable comment. According to your previous comment (comment #6), we added literature in the 1st, 2nd, and 3rd paragraphs of conclusion, red lines.

  1. Conclusion Clarity:The conclusion currently states that "sleep disturbances and PD have a reciprocal relation with each other." This statement is a bit ambiguous and could be made clearer. For instance, the authors could specify whether they mean that PD tends to cause sleep disturbances, or that sleep disturbances might accelerate the progression of PD, or both.

Response 9: Thanks for your precision. In the 4th paragraph of conclusion, this sentence was edited.

Minor suggestions:

  1. Interdisciplinary Approach:The manuscript has a strong focus on the biological aspects of PD and RBD, but it could benefit from a broader, interdisciplinary perspective. For example, the psychological and social impacts of these disorders could be discussed, including how they affect quality of life, mental health, and social relationships. This would provide a more comprehensive understanding of the disorders and could also inform treatment and management strategies.

Response 10: Thanks for your great suggestion. As you point out, this is a biological and neurophysiological study, and we haven't addressed the social aspects of these disorders. Since talking about this requires the help of psychologists or sociologists, and we do not have expertise in this field, and also, this is not the purpose of our study, we think it is better not to mention this point in this review article.

By addressing these points, the manuscript could be significantly enhanced in clarity, comprehensibility, and overall quality.

11- There are some awkward sentences that could be reworded for clarity. For instance, as an example, "Dreams within the REM phase are often intense, obscene, and frightening, and the abnormal motor behaviors occur as a response to the contents of the dreams" could be rewritten as "Dreams during the REM phase, which are often intense, obscene, and frightening, can provoke abnormal motor behaviors as a response." 

Response 11: Thanks for your advice. This sentence was edited (section: 4.1. RBD definition), and English grammar was re-checked.

Round 2

Reviewer 3 Report

Comments and Suggestions for Authors

The authors have done their best in responding to my suggestions so I conclude for acceptance in the present form.

Comments on the Quality of English Language

 Minor editing of English language required.

Author Response

Reviewer 3 comments:

The authors have done their best in responding to my suggestions so I conclude for acceptance in the present form.

Minor editing of English language required.

*Thank you very much for taking the time to review the revised version. English grammar was edited.

Cordially,

Salar Vaseghi (Cognitive Neuroscience Lab, Medicinal Plants Research Center, Institute of Medicinal Plants, ACECR, Karaj, Iran)

P.O. Box: 1419815477

Tel: +982634764017 

E-mail: [email protected] - [email protected]